# Single-Cell RNA Sequencing of the Testis of *Ciona intestinalis* Reveals the Dynamic Transcriptional Profile of Spermatogenesis in Protochordates

**DOI:** 10.3390/cells11243978

**Published:** 2022-12-08

**Authors:** Yanan Li, Xiang Liu, Xianghui Zhang, Hongyan Wang, Jianyang Chen, Jiankai Wei, Yubang Li, Hongxi Chen, Qian Wang, Kaiqiang Liu, Yuyan Liu, Changwei Shao

**Affiliations:** 1College of Fisheries and Life Science, Shanghai Ocean University, Shanghai 201306, China; 2Key Lab of Sustainable Development of Marine Fisheries, Ministry of Agriculture and Rural Affairs, Yellow Sea Fisheries Research Institute, Chinese Academy of Fishery Sciences, Qingdao 266072, China; 3College of Marine Technology and Environment, Dalian Ocean University, Dalian 116023, China; 4Laboratory for Marine Fisheries Science and Food Production Processes, Qingdao National Laboratory for Marine Science and Technology, Qingdao 266071, China; 5BGI Research-Qingdao, BGI, Qingdao 266555, China; 6Qingdao-Europe Advanced Institute for Life Sciences, BGI-Qingdao, Qingdao 266555, China; 7Sars-Fang Centre, MoE Key Laboratory of Marine Genetics and Breeding, College of Marine Life Sciences, Ocean University of China, Qingdao 266003, China; 8College of Life Sciences, Shandong Normal University, Jinan 250014, China

**Keywords:** spermatogenesis, scRNA-seq, *Ciona intestinalis*, protochordates

## Abstract

Spermatogenesis is a complex and continuous process of germ-cell differentiation. This complex process is regulated by many factors, of which gene regulation in spermatogenic cells plays a decisive role. Spermatogenesis has been widely studied in vertebrates, but little is known about spermatogenesis in protochordates. Here, for the first time, we performed single-cell RNA sequencing (scRNA-seq) on 6832 germ cells from the testis of adult *Ciona intestinalis*. We identified six germ cell populations and revealed dynamic gene expression as well as transcriptional regulation during spermatogenesis. In particular, we identified four spermatocyte subtypes and key genes involved in meiosis in *C. intestinalis*. There were remarkable similarities and differences in gene expression during spermatogenesis between *C. intestinalis* and two other vertebrates (Chinese tongue sole and human). We identified many spermatogenic-cell-specific genes with functions that need to be verified. These findings will help to further improve research on spermatogenesis in chordates.

## 1. Introduction

Spermatogenesis is a complex cellular transformational biological process that generates male haploid germ cells from diploid spermatogonial stem cells. In vertebrates, spermatogenesis is divided into three stages: the proliferation and differentiation of spermatogonia, meiosis of spermatocytes, and spermiogenesis. The fate of spermatogonia can follow two paths: self-renewal through mitosis and differentiation into spermatozoa through meiosis, which gives rise to undifferentiated spermatogonia including stem cells and differentiated or differentiating spermatogonia [1]. The balanced and orderly transition between spermatogonial stem cell mitosis and meiosis ensures the continuous generation of male gametes. The last S phase of germ cells occurs in preleptotene spermatocytes before entering the very long prophase of the first meiosis. Secondary spermatocytes formed during the first meiosis immediately enter the second meiotic cell cycle and skip DNA synthesis, so they are very short-lived. In the subsequent M phase, haploids are isolated from the spermatid nuclei. Ultimately, spermatids undergo a major physical and structural reorganization to transform into sperm with flagella. This complex process of cell differentiation is regulated by many factors, among which gene regulation in spermatogenic cells plays a decisive role. Genes known to be related to spermatogenesis include cyclin genes, protooncogenes, cytoskeleton genes, nuclear protein transformation genes, and apoptosis-related genes, etc. [2]. Spermatogenesis is a relatively conserved process in vertebrates [3]. However, it is still unclear whether protochordates have retained the basic transcriptional features of spermatogenesis throughout evolution.

Meiosis is the decisive event in spermatogenesis, and it is also what ensures species’ reproduction, stable chromosome numbers, and continuous evolution to adapt to environmental changes. The first meiotic prophase is the longest stage of meiosis. Spermatocytes in the first meiotic prophase are subdivided into leptotene, zygotene, pachytene, and diplotene spermatocytes according to their chromosome morphology in different stages [4]. During the first meiotic prophase, paternal and maternal chromosomes begin pairing. This progress involves a group of meiosis-specific proteins, the synaptonemal complex proteins (*SYCP1*, *SYCP3*, etc.), which mediate the tight association between homologous chromosomes and facilitate their subsequent repair in the event of meiotic crossover [5]. From single-cell yeast to mammals, the key biological processes of meiosis, including homologous chromosome pairing, recombination exchange, and separation, are highly conserved [6]. For example, the meiotic recombinases *Rad51* and *Dmc1* in yeast, higher plants, and mammals are very similar in molecular organization and function across different taxa. They are involved in identical processes at the molecular level: the formation of DNA heteroduplexes and Holiday structures and their processing during meiotic DNA recombination [7]. Nevertheless, the molecular mechanisms regulating meiosis also differ significantly among different organisms. Most of the key genes that regulate meiosis in yeast, *Drosophila*, and nematodes have no homologous genes in mammals [8,9]. At present, the meiotic process of protochordates, which are in a transitional evolutionary position, has not been thoroughly studied.

*C. intestinalis* are hermaphroditic urochordate animals that can be found in the evolutionary transition from invertebrates to vertebrates. The unique phylogenetic status of this animal provides attractive and useful targets for a wide range of biological research topics, including developmental biology, evolutionary biology, endocrine science, and neuroscience. *C. intestinalis* has outstanding advantages as a model organism, especially its whole genome sequence, various ESTs, and microarray analysis data, which can be used for various gene model predictions, homology searches, and comprehensive comparisons with the genomes and transcriptomes of other species [10,11,12]. Previously, EST analysis revealed the specific expression of a large number of genes in the *C. intestinalis* testis [13]. In addition, *Ci-DEAD1* has been shown to be a germline-specific gene expressed in *C. intestinalis* [14]. These findings indicate that *C. intestinalis* has significant potential as a model organism for studying the spermatogenesis of urochordates and the evolution of spermatogenesis throughout chordates. With the development of sequencing technology, more convenient and accurate methods can now be used to evaluate the transcriptional profiles of different spermatogenic cells in the *C. intestinalis* testis.

In this study, we used scRNA-seq to reveal the transition of germ-cell types and transcriptional signatures during spermatogenesis in *C. intestinalis*. Furthermore, we described the four stages and key genes during meiosis in detail. Our analysis of *C. intestinalis* spermatogenesis is complemented with comparative analyses performed on two other vertebrate species (Chinese tongue sole and human). These studies provide preliminary insights into spermatogenesis in urochordates.

## 2. Materials and Methods

### 2.1. Dissociation of Testis Cells in C. intestinalis

The testis tissue of adult *C. intestinalis* was carefully dissected, placed in a sterile culture dish, rinsed with 5 mL DMEM+2% NaCl for 3–5 min, and evenly chopped with a blade to form a homogenate. The tissue homogenate was transferred to a 5 mL centrifuge tube containing digestive enzyme solution (ratio of trypsin to collagenase: 5 to 1), mixed upside down, sealed, and then placed into a 30 °C water bath for digestion for approximately 5–10 min. During digestion, the centrifuge tube was removed every 1 min and turned upside down to prevent tissue adhesion and allow even contact between the tissue and digestive enzymes. With a 40 μm cell sieve, the filtered cell suspension was transferred to a 15 mL centrifuge tube and then approximately 7 mL DMEM+2% NaCl was added. A 4 °C centrifuge with a horizontal rotor at a speed of 500× *g* was used for 3–5 min. After removing the supernatant, 7 mL of PBS+2% NaCl was added, centrifuged at 400× *g* for 3–5 min, and washed twice. The supernatant suspension was removed one last time, 0.1% BSA in PBS+2% NaCl was added, and the cells were resuspended. Finally, single-cell suspension of ascidian gonads was mixed with trypan blue, cell activity was observed (>80%), and cell concentration (1000–2000 cells per 1 μL) was calculated using a cell counter.

### 2.2. scRNA-Seq Library Preparation and Sequencing

We used the DNBelab C Series Single Cell RNA Library Preparation Kit (MGI Tech Co., Ltd., Qingdao, China) based on droplet microfluidics technology for library construction. Briefly, cells were prepared as droplets, in which cell lysis and mRNA capture were performed using the DNBelab C4 portable single-cell system. Single-cell microdroplets were recovered by using the emulsion breaking recovery system, after which mRNA captured by magnetic beads was synthesized into cDNA; 14 cycles were used for cDNA amplification. Finally, the cDNA product was used for 11 cycles of library amplification. Ten nanograms of the digested product was then collected for sequencing on the MGISEQ 2000 platform.

### 2.3. scRNA-Seq Data Processing

The cell barcodes and UMIs of the raw data were integrated into fastq files by the parse option in PISA (v1.10.2), and quality control was performed by FastQC (v0.11.3). Reads were aligned to the reference genome from NCBI using STAR (v2.7.9a). The sam2bam and anno options of PISA were used successively to convert the barcodes to tags in the BAM files and annotate reads and add annotation tags in the BAM files. Sambamba (v0.7.0) was used to sort BAM files. The cell versus gene UMI count matrix was generated using PISA.

### 2.4. H&E Staining

Testes from adult *C. intestinalis* were fixed with 4% paraformaldehyde overnight at 4 °C. Then, the fixed samples were embedded in OCT after sugar dehydration, sectioned at 10 μm thickness, and stained with hematoxylin and eosin.

### 2.5. Gene Function Enrichment Analysis

We used Metascape (https://metascape.org/gp/index.html, accessed on 25 May 2022) for gene function enrichment analysis. Genes of *C. intestinalis* were transformed into *H. sapiens* orthologous genes for enrichment analysis.

### 2.6. Pseudotime Trajectory Analysis

The Monocle3 (v1.2.9) R package was used for pseudotime analysis [15]. The integrated data, dimension reduction, and clustering information were imported from Seurat to the Monocle3 package. The function “learn_graph” was used to learn the trajectory through the inverse graph embedding algorithm of Monole3. Subsequently, cells were ordered in pseudotime using the function “order_cells”.

### 2.7. Coexpression Network Construction and Analysis

We searched for hub genes in spermatocytes by WGCNA (v1.71) [16]. To prevent the influence of sparsity and the noise from single cell matrices, we constructed the mean gene expression per 20 cells in all cell types as pseudocell data and retained only 5000 highly variable genes for subsequent analysis. Afterward, to construct the weighted gene coexpression network (WGCN) with scale-free topology, different values of soft thresholding power β were assessed for the network topology analysis, and the value of 6 was selected. The topological overlap measure (TOM) was used to identify modules; this method investigates the similarity between gene pairs based on the number of shared neighbors in the generated coexpression network. Modules in the WGCN were depicted in different colors. The relationships between the detected modules were shown as module eigengenes from the first principal component of the expression values in modules. Finally, hub genes in spermatocyte-related modules were selected from the top 100 genes with the highest intramodular connectivity (sum of the in-module edge weights). The constructed WGCN was exported to Cytoscape for visualization [17].

### 2.8. Cross-Species Orthologous Gene Pairs Construction

1-1 orthologs. By using Orthofinder (v2.3.7) with default parameters to cluster gene families, single copy orthologous genes were identified from the genomes of the two species. Single-copy genes were extracted from the output file of Orthofinder (4821 *C. intestinalis*–*C. semilaevis* pairs, 9051 *C. semilaevis*–*H. sapiens* pairs, and 5266 *C. intestinalis*–*H. sapiens* pairs).

1-1-1 orthologs. Here, 4273 *C. intestinalis*–*C. semilaevis*–*H. sapiens* groups were found using the methods described above.

Other orthologs. Other orthologous genes not in the 1-1-1 orthologous list between species, such as ‘1-many’ and ‘many-many’.

Species-specific genes. Those genes that had no ortholog annotation were considered species-specific genes.

### 2.9. Cell-Type Similarity Comparison

We use MetaNeighbor [18] to calculate the degree of similarity of three species’ cell types. Similar to the previous analysis of cell-type evolution [19], a MetaNeighbor analysis was performed based on the pseudocell data. The similarity of cell types was measured by the average area under the receiver operator characteristic curve (AUROC) score.

To compare *C. intestinalis* cell types with those of the other species, we used the same method as in the previous stone coral cell atlas [20]. We generated paired datasets of orthologous gene expression. When a species has multiple pairs of orthologous genes in another species, we can find 1-many or many-many orthologs by copying the gene entries from one species. This resulted in 11,494 *C. intestinalis*–*C. semilaevis* pairs, 17,902 *C. semilaevis*–*H. sapiens* pairs, and 10,575 *C. intestinalis*–*H. sapiens* pairs. Similarity between cell types was measured by Kullback–Leibler divergence (KLD). Links between pairs of cell types with the smallest KLD (0.85 quantile) are shown in the circos plot, generated using the R package circlize. For resulting groups of cell types, we compared the expression of shared orthologs (FC > 1.3) to their expression in other cell types; significance was calculated using paired Wilcoxon test between in- and outgroup cell types.

## 3. Results

### 3.1. Single-Cell RNA Sequencing Identifies Germ-Cell Types Present within C. intestinalis

We extracted testis tissue from the gonads of adult *C. intestinalis*. At this stage, the testis contains spermatogenic cells at various stages (Figure 1A). We constructed 9 sequencing libraries using the iDrop system and cells that passed the standard quality control (QC) dataset filters were retained. After cell clustering using t-distributed stochastic neighbor embedding (tSNE) analysis, we focused only six germ-cell types (6832 cells in total) for subsequent analysis based on the combined expression of the germline-specific marker genes (*ddx4, sycp3* and *spag6*) (Figure 1B,C) [14,21]. On average, each cell had 4812 unique molecular identifiers (UMIs) and 1273 genes (Appendix A). All clusters showed only minor changes based on their batch/experiment (Appendix A).

We identified these six germ-cell types in different states through differentially expressed genes (DEGs), including undifferentiated spermatogonia (Undiff SPG), differenting spermatogonia (Diff.ing SPG), differentiated spermatogonia (Diff.ed SPG), early primary spermatocytes (Early primary SPC), late primary spermatocytes (Late primary SPC) and sperm (Appendix A). Undiff SPG was identified by high expression of spermatogonia characteristic genes (*piwil1*, *ddx4*, *fgfr3*, etc.) (Figure 1D,E) [14,22]. Diff.ing SPG highly expressed spermatogonia differentiation-related genes *calr* and *supt16h* (Figure 1D) [23]. *Topbp1*, which is essential for G1/S transition [24], was more prominently expressed in Diff.ed SPG (Figure 1D). Early primary SPC mainly expressed early meiotic genes (*dmc1*, *ccna2*, *sycp3*, etc.) (Figure 1D,E) [21,25,26]. Subsequently, *sycp1*, marker of meiotic-specific synapsis, and *aurka*, cell cycle kinase, were mainly expressed in Late primary SPC (Figure 1D,E) [27]. Sperm was identified by its marker genes (*spag6*, *tekt1*, etc.) [28] (Figure 1D,E). We also identified the authenticity of marker genes based on their conserved domains (Appendix A). We further performed enrichment analysis of DEGs for each cell type using the Metascape database. The results showed that the “piRNA metabolic process”, “cell cycle”, and “G1/S Transition” were enriched in Undiff SPG, Diff.ing SPG, and Diff.ed SPG, respectively; the “chromosome organization involved in meiotic cell cycle” and “cilium organization” were enriched in Early primary SPC and Late primary SPC; the “sperm mobility” and “sperm development” were enriched in sperm (Figure 1F). Taken together, these genes and pathways reflect the sequence of spermatogenesis.

### 3.2. Pseudotime and Clustering Analysis Revealed Dynamic Gene-Expression Patterns during Spermatogenesis

We used Monocle3 to model the temporal sequence of male germ-cell development in *C. intestinalis*. Reclustering was performed using uniform manifold approximation and projection (UMAP) analysis due to the applicability of the software (v1.2.9) (Figure 2A). The pseudo-developmental trajectory from undifferentiated spermatogonia to spermatozoa is consistent with prior knowledge (Figure 2B) [29].

Cell cycle status was assessed by analyzing G1/S and G2/M phase-specific genes. In Undiff SPG, the cell proliferation state was relatively inactive (Appendix A). After the initiation of differentiation, both the G1/S and G2/M genes were active. G1/S phase-related genes were enriched in Early primary SPC. Subsequently, decreased expression of G1/S phase-specific genes and increased expression of G2/M phase-specific genes were observed in Late primary SPC. Finally, sperm exit the cell cycle (Appendix A). We also found that germ cells differentiated from spermatogonia to Early primary SPC with the high level of expression genes (*mcm5*, *cdc6*, *nasp*, etc.), which were involved in the control of G1/S progression (Appendix A) [30,31]. In contrast, Late primary SPC were enriched in the G2/M phase drivers (*nek2*, *ube2c*, *ccnb2*, etc.) (Appendix A) [32].

In addition, we analyzed the global expression pattern of genes during spermatogenesis. We identified 1777 genes that varied with the pseudo-developmental order of germ cells, revealing four distinct gene groups (Figure 2C). As expected, the term “mitotic cell cycle” was enriched in group1, “meiotic cell cycle” in group2, and “cilium organization” in group4 (Figure 2D). This result shows the transition of spermatogenesis from mitosis to meiosis and the eventual generation of flagellar motility-dependent sperm. Furthermore, we also found some other enriched terms, such as “regulation of MAPK cascade” in group1, “ncRNA metabolic process” in group2, and “extension of telomeres” in group3, which play a broad role in spermatogenesis [33,34,35] (Figure 2D). The spermatogenesis of *C. intestinalis* is also regulated by some transcription factors. We showed that the transcription factors that are differentially expressed across stages (Figure 2E). The expression of c*i-htf-1* decreased gradually during spermatogenesis. *Lag1-like1* was up-regulated from Undiff SPG to Diff.ing SPG. The CREB/ATF family members *creb/atf-c* and high-mobility histones *hmg1/2* were up-regulated from the Diff.ing SPG until they differentiated into Early primary SPC [36]. The expression of *soxh,* which is involved in male germ-cell development, gradually increased after entering Early primary SPC. *Rfx4* was highly expressed at the end of spermatogenesis (Figure 2E). Our results provide a reference for the detailed functional study of these transcription factors in spermatogenesis.

### 3.3. Analysis of Spermatocyte Subsets Reveals Transcriptional Patterns and Key Factors during Meiotic Transition

To further explore spermatocyte heterogeneity, we singled out two spermatocyte clusters and reclustered these meiotic cells into four sub-clusters (Figure 3A). Distinct shifts in gene expression corresponded to key cell-state transitions: from preleptotene (Pre-Lep) to leptotene (Lep-SPC), then to pachytene (P-SPC), and finally to diplotene (D-SPC). As cells develop through each subcluster, the transcription changed gradually (Figure 3B). For instance, genes (*ccna2*, *mcm4*, *pds5a/b*, *rec8*, etc.) involved in DNA replication and the meiotic cell cycle were prominently characterized in Pre-Lep, with decreased expression in Lep-SPC. Genes (*sycp1*, *nme8*, *mns1*, *spag6*, etc.) involved in male gamete generation were subsequently upregulated in P-SPC and were most pronounced in D-SPC. In the D-SPC phase, genes (*plk1*, *cfap157*, etc.) that function in microtubule bundle formation began to be expressed (Figure 3B). In addition, the state of the cell cycle in different stages was also different, and most of the cells in the Pre-Lep were in the S phase (Figure 3C and Appendix A).

To define candidate key genes and networks associated with meiotic cell developmental stages, we performed weighted gene coexpression network analysis (WGCNA). The gene-clustering tree generated based on the degree of expression association between genes produced five gene modules (Figure 3D). Among them, the blue module was significantly related to late meiotic cells (P-SPC and D-SPC) and sperm Figure 3E and Appendix A), and we showed high expression of its genes in late meiotic cells (Appendix A). The brown module had a strong correlation with spermatocytes at all stages (Figure 3E and Appendix A), and most genes showed relatively extensive expression (Appendix A). We identified interactions between the top 100 hub genes in the blue module, with top 12 genes highlighted (Figure 3F). Some of these genes (*spag6*, *qrich2*, *c10orf63*, and *efhc2*) were involved in the development of spermatids during spermatogenesis (Figure 3F), reflecting the transition of cells to spermatids at the late stage of meiosis [37,38]. The brown module contained fewer genes, and the analysis revealed the importance of the testis-specific tubulin alpha chain 2C (*tuba2c*) genes in *C. intestinalis* spermatocytes (Figure 3G). Furthermore, our analysis also found additional factors that deserve further exploration (*ak7*, *pacrg*, *ttc25*, *ckmt2*, *fbxo36*, etc.) (Figure 3F,G). For example, male mice lacking *Pacrg* have an infertile reproductive phenotype, and homozygous missense mutations in *AK7* in humans lead to sperm deformities [39,40]. Thus, our results confirm the role of some known factors in the meiotic process of *C. intestinalis*, while also providing candidate markers for future analyses.

### 3.4. Conservation and Differential Characteristics of Spermatogenesis in Chordate Species

To explore cell-type conservation and specific transcriptional signatures during spermatogenesis in species within the chordates, we compared the testis germ-cell data of *C. intestinalis* (vase tunicate; urochordata) with the available scRNA-seq datasets of *C. semilaevis* (Chinese tongue sole; vertebrate) and *H. sapiens* (human; vertebrate).

We first processed the obtained data using the same analysis method and named cells separately, according to the annotations of their corresponding references [41,42]. Cell-type expression matrices were then integrated using the 1-1-1 orthologous genes found in the three species. The comparative analysis of the germ-cell types of the three species showed that germ cells at different stages had high transcriptional similarity (Figure 4A).

Further, we delved into cell-type comparisons between pairs of species. The results showed that *C. intestinalis* had multiple germ-cell similarities to *C. semilaevis* and *H. sapiens,* respectively, and had specific coexpression of orthologous genes (Figure 4B,C). For instance, the Diff.ing SPG of *C. intestinalis* were more similar to the Pre-Lep of *C. semilaevis* as well as to the Early primary SPC of *H. sapiens* (Figure 4B). The common feature was the high expression of 20 orthologs, including most cell-cycle-related genes, such as the chromosome structure maintenance (SMC) family members *SMC1A*, *SMC2*, *SMC3*, and *SMC4* (Figure 4C). Early primary SPC of *C. intestinalis* were more similar to those of *C. semilaevis*, with a total of 23 expressed orthologous genes, including the known meiotic marker *SYCP3* (Figure 4B,C). There were many orthologous genes shared by the sperms of *C. intestinalis* and the spermatids of *H. sapiens* (Figure 4C). However, we also observed some many-to-many germ-cell similarities across species, a pattern that illustrates the evolution of diversity in germ-cell lineages.

Subsequently, we turned to the analysis of gene-expression differences in the process of spermatogenesis in *C. intestinalis*, *C. semilaevis*, and *H. sapiens*. A total of 2456 differentially expressed 1-1-1 orthologous genes were identified, which may play unique roles only in the spermatogenesis of specific species (Figure 5A). Enrichment analysis of these species-specific gene sets also revealed predominantly species-specific functional terms, suggesting that even though they are orthologs, many of these genes may play distinct roles in each species (Figure 5B). Finally, we assessed the diversity of spermatogenesis in the three species by comparing the expression levels of orthologs and species-specific genes. Interestingly, we found that the overall expression of 1-1-1 orthologs increased in *C. intestinalis* as cells entered meiosis. In turn, the expression level of species-specific genes decreased. This finding reflects that the cell types of the later stage of spermatogenesis in *C. intestinalis* and the other two species are more conserved. In addition, the overall expression of species-specific genes was particularly prominent in *C. intestinalis* and *H. sapiens*, and second only to other orthologs in *C. semilaevis,* implying that, in addition to the evolution of conserved features, independent specialization was more pronounced during spermatogenesis (Figure 5C).

## 4. Discussion

Spermatogenesis involves the transcriptional changes of male germ cells at different differentiation stages. Due to its high throughput and accuracy, scRNA-seq technology has been applied to the study of spermatogenesis in mammals and fish [23,41,43,44,45]. However, as a close relative of vertebrates, the gene-expression profile of spermatogenesis in protochordates has not been verified. Here, we present basic scRNA-seq data from all germ cells in the adult testis of *C. intestinalis*, complemented with computational analysis, and we provide new insights into the regulation of chordate gametogenesis. 

In this study, we identified major germ-cell populations, including spermatogonia (SPG), spermatocytes (SPC), and sperm, with known marker genes. Notably, the typical spermatogonia markers in vertebrates (*KIT*, *UTF1*, *ID4*, *SOHLH1*, *UCHL1*, *STRA8*, *DND1*, *GFRA1*, *POU5F1/OCT4*, etc.) do not have homologs in the genome of *C. intestinalis* at present. This may be because during spermatogenesis in different species, the mitotic stage differs due to the different generations of spermatogonia, and the meiotic stage and spermatogenesis stage are more conserved [46]. Several new marker genes were identified for germ cells in a study on the transcriptome of adult testis in orange-spotted grouper [23]. For example, *calr* was used as spermatogonia marker genes in the present study. The *piwi* gene is known to be identified as a marker of hemoblasts in ascidians [47], and the *piwil1* gene was predominantly expressed in spermatogonia in our data. It has been found that the germ cells in the gonads of *Botryllus primigenus* can be developed from the PIWI^+^ hemoblasts in coelomic tissue [48]. The differences between hemoblasts and spermatogonia in *C. intestinalis* require further study. In addition, we also found some unique gene-expression profiles in *C. intestinalis* sperm, including calaxin, which can control the correct swimming of sperm released by ascidians in seawater toward the egg [49], and s(sperm)-Themis-A and s(sperm)-Themis-B, which are responsible for the hermaphroditic self-sterility system of ascidians (Figure 1B) [50]. Moreover, ascidians also naturally have the ability to control their reproductive cycle and can reproduce more or less in moderation, according to need [51]. Our data on the spermatogenesis of *C. intestinalis* lay a foundation for studying these peculiar mechanisms in its germ cells.

Spermatogenesis and maturation are closely related to signal pathway transduction. The piwi/piRNA pathway plays a role in transposon silencing in germ lines. Mutations in most piRNA pathway factors in mice lead to meiotic arrest during spermatogenesis [52]. Endoplasmic reticulum (ER) stress is associated with male reproduction and infertility in animal models. ER stress could be a novel signaling pathway regulating male reproductive cellular apoptosis [53]. *C. intestinalis* spermatogonia DEGs were enriched in terms such as “piRNA metabolic process” and “response to ER stress”, reflecting the conservation of these biological processes in chordate spermatogenesis. We found that most genes related to spermiogenesis and sperm function have been turned on in late primary spermatocytes, such as the enrichment of “cilium organization”. This indicates that the transcriptional transition from meiosis to post-meiosis takes place very early in the prophase of meiosis, which is consistent with findings in mice [54].

The results of pseudotime analysis showed that *C. intestinalis* spermatogenesis followed a continuous developmental trajectory, and the initial step was presumed to be the balance of self-renewal and differentiation of spermatogonia. This process involves the transition from mitotic to meiotic cell cycles and contains multiple conserved pathways involved in the spermatogenesis process, as described in the results. In addition, we detected differentially expressed transcription factors during spermatogenesis, such as *lag1-like1*, *ci-htf-1*, *hmg1/2*, *soxh*, and *rfx4*. *SOXH* is a member of the SOX gene family, which encodes an ortholog of *SOX30*. *SOX30* is a key spermatogenesis regulator required for normal meiosis in the vertebrate testis. The high expression of *soxh* in the testis germ cells of *C. intestinalis* after meiosis indicates that the *soxh* gene can be used as an important regulator of conserved male gonadal function in chordates [55].

Meiosis is the key event of spermatogenesis. The correct completion of meiosis ensures the stable transmission of genetic information and the diversity of species. This gonad-specific cell division begins with primary spermatocytes, which are fully committed to meiosis and participate in gene recombination at this stage. For the first time, we identified four primary spermatocyte subpopulations in *C. intestinalis*, including preleptotene, leptotene, pachytene, and diplotene spermatocytes. The secondary spermatocytes produced by the first meiosis rapidly enter the second meiosis, so we did not capture the secondary spermatocytes. In the first meiosis, homologous chromosomes undergo a series of complex events, including pairing, association, recombination, and separation [56]. In mammals, *REC8* begins to appear in the centromere and adjacent chromosome arms at the premeiotic S phase, and it regulates sister chromatid condensation and recombination between homologous chromosomes [57]. PDS5 protein ensures the integrity of telomeres and their attachment to the nuclear membrane [58]. *DMC1* participates in double-strand break (DSB) repair and exchange [59]. During pachytene, *SYCP1* is involved in homologous chromosome recombination. *MNS1* is a nuclear skeletal protein that regulates nuclear morphology during meiosis and is also required for sperm flagella assembly [60]. Our data demonstrate the expression of these genes in the spermatocytes of *C. intestinalis*. In addition, some additional factors worthy of further exploration were also found, as shown in Figure 3F,G.

Our comprehensive analysis of spermatogenic cells from three chordates highlights the similarities and differences in various features during spermatogenesis. Cell types from different stage of spermatogenesis showed a high level of transcriptional similarity. This indicates that even species with different reproductive modes have conserved transcriptional features of spermatogenesis. Naturally, there is also obvious diversity among different species. This can be seen by the differential expression of orthologous genes and the prominent contribution of species-specific genes. Among them, the 1-1-1 orthologs in the three species of *C. intestinalis* seem to be more prominent after the beginning of meiosis, indicating that the gene-expression networks of the three species in meiosis and spermiogenesis are more conserved. This is related to the absence of most typical spermatogonia marker genes from our data in the previous analyses.

In conclusion, for the first time, we identified six germ-cell types representing the testis of *C. intestinalis* by scRNA-seq and revealed the dynamic gene expression and transcriptional regulation during spermatogenesis. In particular, we identified four spermatocyte subtypes and key genes involved in meiosis in *C. intestinalis*. Moreover, we explored the evolutionary conservation and diversity of spermatogenesis in different chordates. These findings are of great significance for the in-depth exploration of the developmental mechanism of urochordata germ cells and will help to further improve research on chordate spermatogenesis.

## Figures and Tables

**Figure 1 cells-11-03978-f001:**
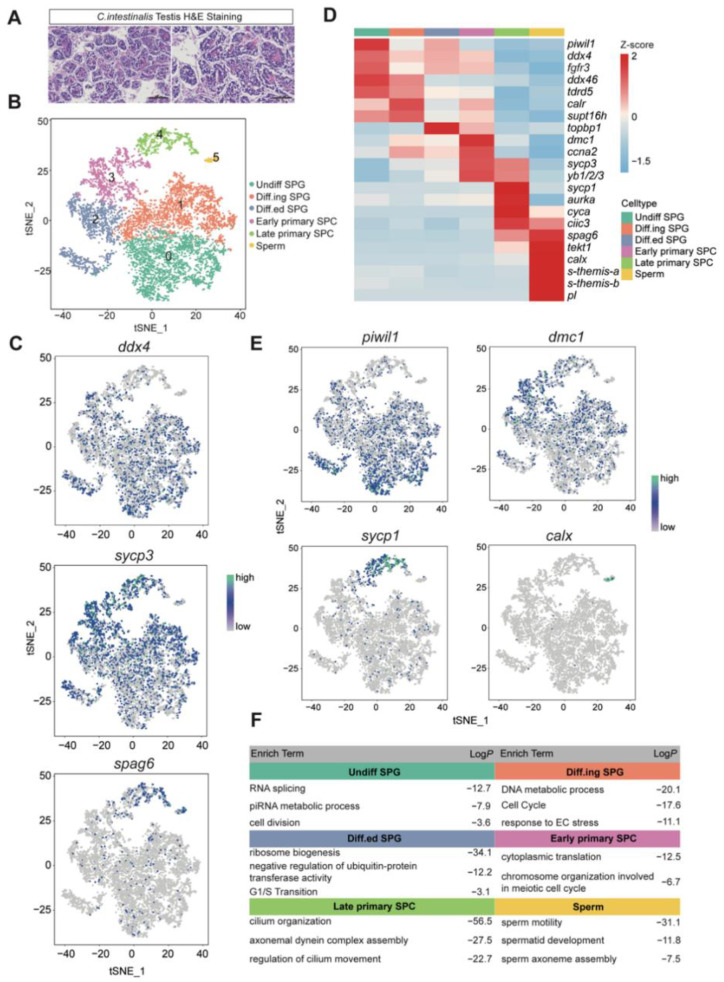
Single-cell transcriptome analysis identified germ cells of *C. intestinalis testis*. (**A**) Histological and morphology of *C. intestinalis* testes by H&E staining. Left: scale bar = 100 µm; right: scale bar = 50 µm. (**B**) t-SNE and clustering analysis of scRNA-seq data from *C. intestinalis* testis germ cells (n = 6832). Different cell types are shown in distinct colors. The 6 cluster identities were assigned based on marker gene expression shown in (**C**–**E**). Undiff SPG: undifferentiated spermatogonia; Diff.ing SPG: differentiating spermatogonia; Diff.ed SPG: differentiated SPG; SPC: spermatocyte. (**C**) t-SNE plot showing the expression of genes *ddx4*, *sycp3* and *spag6*. (**D**) Heatmap showing the expression of marker genes in different cell types. Differential gene-expression levels utilize a Z score, which represents the variance from the mean, as defined in the color key in the top-right corner. (**E**) t-SNE plot showing the expression of major marker genes. (**F**) Enriched terms for DEGs are shown for germ-cell types (*p* values are shown).

**Figure 2 cells-11-03978-f002:**
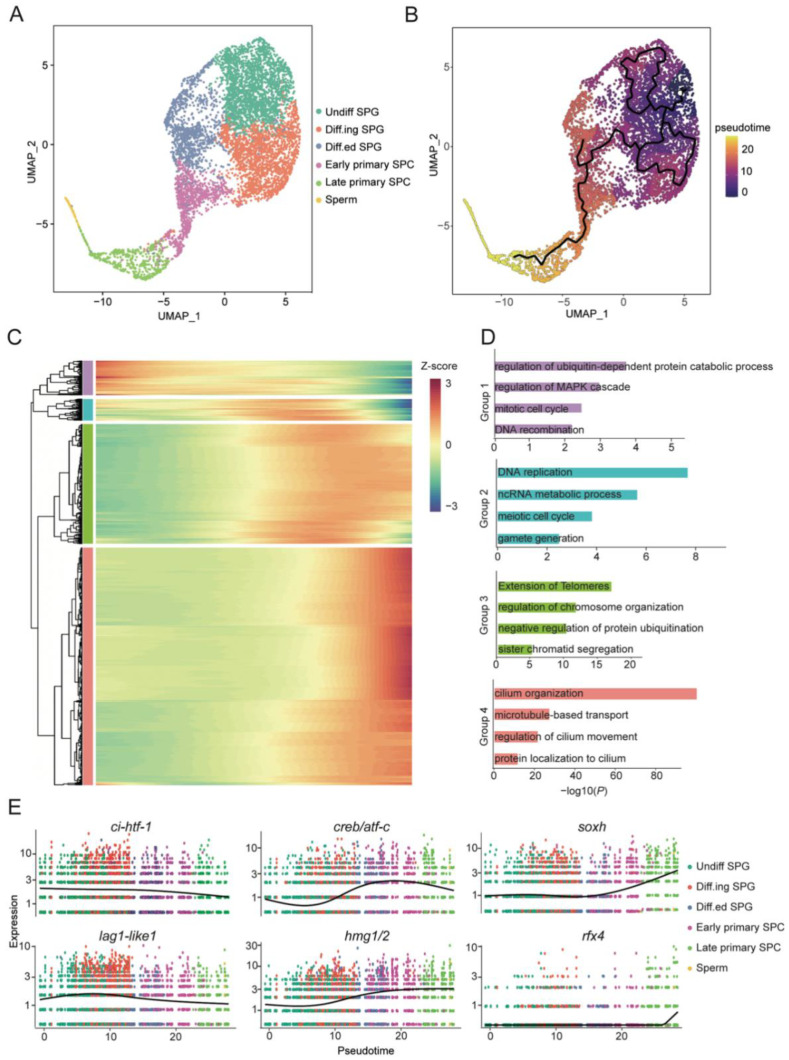
Gene-expression dynamics during spermatogenesis in *C. intestinalis*. (**A**) UMAP plot of 6 germ cell clusters. (**B**) Germ-cell pseudotime trajectories analyzed using Monocle3. Different colors indicate the differentiation degree of cell type (darker colors indicate lower differentiation degree). (**C**) K-means clustering of genes exhibiting differential expression (n = 1777) across germ-cell populations. (**D**) Enriched terms for each gene group in (**C**). (**E**) Expression patterns of transcription factors during spermatogenesis.

**Figure 3 cells-11-03978-f003:**
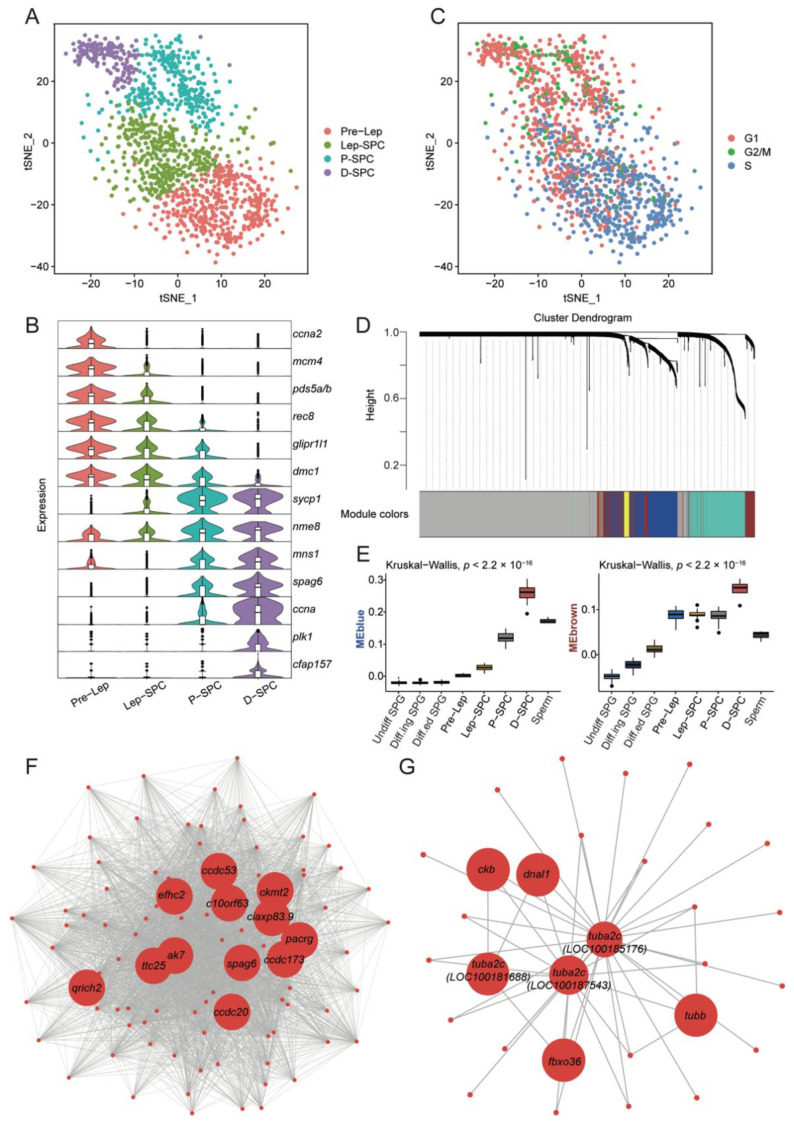
Identification of four meiotic cell populations in *C. intestinalis*. (**A**) t-SNE of spermatocyte re-clustering. Different colors correspond to 4 different stages of spermatocyte subsets. Pre-LEP: preleptotene spermatocytes; Lep-SPC: leptotene spermatocytes; P-SPC: pachytene spermatocytes; D-SPC: diplotene spermatocytes. (**B**) Vlnplot of the normalized expression of meiotic marker genes. (**C**) t-SNE plot showing the cell cycle status of spermatocytes. (**D**) The clustering dendrogram of the weighted gene coexpression network. The resulting modules are depicted in different colors: blue, turquoise, yellow, brown, and gray. (**E**) Boxplot showing the correlation of the blue module and brown module with all germ-cell types. (**F**) Interactions of the top 100 genes in the blue module, with the top 12 hub genes highlighted. (**G**) Interactions of all genes in brown module, with the top 7 hub genes highlighted.

**Figure 4 cells-11-03978-f004:**
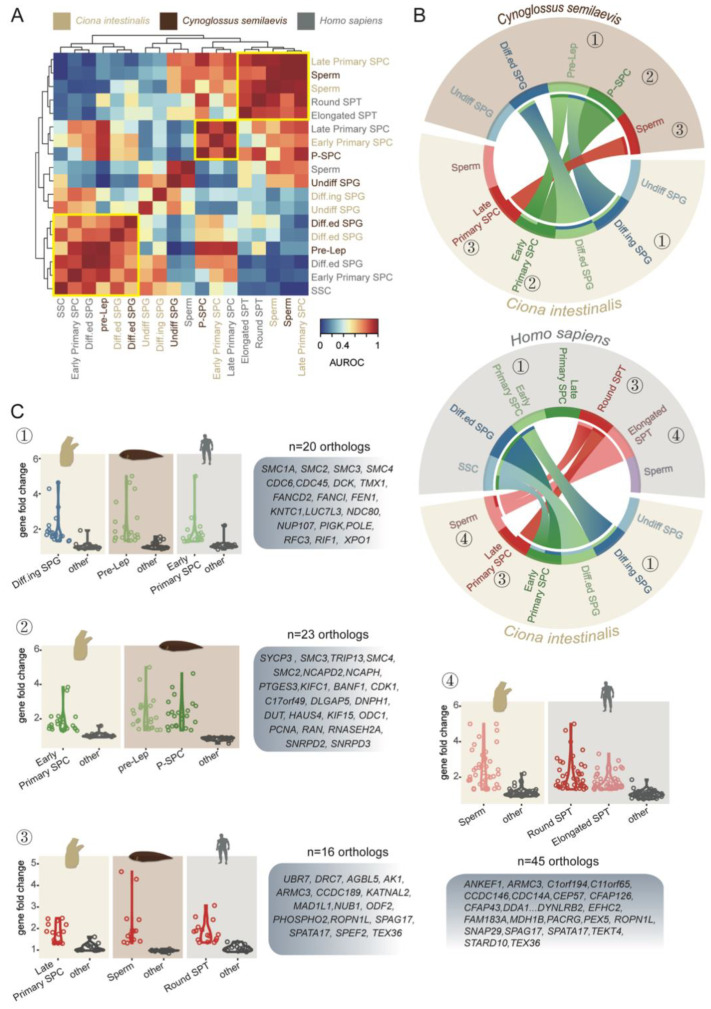
Conserved evolution of spermatogenesis in three chordates. (**A**) Heatmap showing ‘one versus best MetaNeighbor’ scores for germ-cell types. AUROCs are computed between the two closest neighbors in the test dataset, where closer neighbors will have higher scores. SSC: spermatogonial stem cell; Round SPT: round spermatid; Elongated SPT: elongated spermatid; other abbreviations are the same as those in *C. intestinalis*. (**B**) Cross-species pairwise cell-type similarities between *C. semilaevis* and *C. intestinalis* and between *H. sapiens* and *C. intestinalis* based on Kullback–Leibler divergence (KLD). The top 15% of the values are indicated as arches connecting cell types (with a width proportional to the KLD similarity). (**C**) Examples of highly similar germ-cell types across species. Violin plots show the normalized expression of the top shared orthologous genes (FC > 1.3) in each cell type compared to the average expression of the same genes in all other cell types.

**Figure 5 cells-11-03978-f005:**
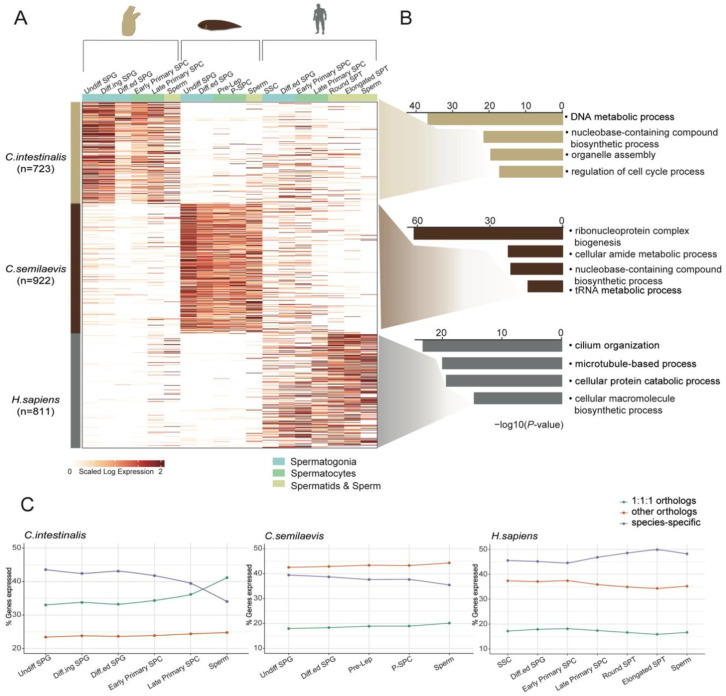
Species-specific evolutionary features of chordate spermatogenesis. (**A**) Heatmap showing the expression of differential gene orthologs among germ cells of the three species. (**B**) GO terms for the genes described in Figure 5A. (**C**) Line plot depicting the percentage of gene classes that are expressed in each species over the course of spermatogenesis, e.g., 1-1-1 orthologous genes (green), other orthologs (orange), and species-specific genes (blue).

## Data Availability

Data of this project can be accessed after an approval application to the China National Genebank (CNGB, https://db.cngb.org/cnsa/, accessed on 24 November 2022). Please refer to https://db.cngb.org/ (accessed on 24 November 2022) or email CNGBdb@cngb.org for detailed application guidance. The accession code CNP0003562 should be included in the application.

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
