# Peer review of "Single-Cell RNA Sequencing of the Testis of Ciona intestinalis Reveals the Dynamic Transcriptional Profile of Spermatogenesis in Protochordates"

_cells, 2022, doi:10.3390/cells11243978_

Round 1

Reviewer 1 Report

In the current paper, Li et al., studied the transcriptional profile of spermatogenesis in C. intestinalis. They investigated the gene expression profile of spermatogenesis in germ cells in the adult testis of the urochordate. They reviewed previous work in good depth and used relevant references with limited self-citation. The unique properties of this organism model have been highlighted in the introduction. In addition, this paper is considered the first of its kind that identified six germ cell types in the testis of C. intestinalis and differentially expressed genes and their dynamics within these cells.

Li et al., utilized single-cell RNA sequencing which is a high throughput and unbiased approach to identify germ cell types present within the C.intestinalis They also utilized computational and comparative analysis of other species to further support their findings.

The results were illustrated clearly and raw data were provided as supplementary files. The discussion was parallel with the obtained results. All in all, this manuscript showed original work and is worth publication without any modification.

Author Response

Thanks for your comment!

Reviewer 2 Report

Presented article named "Single-cell RNA Sequencing of the testis of Ciona intestinalis 2 reveals the dynamic transcriptional profile of spermatogenesis 3 in protochordate" exerts very nice scientific representations, however, it needs further revision as followed 

1. Abstract can be modified in appropriate form as per the update of the MS.

2. In the Introduction part spermatogenesis and spermatogonia can be described more clearly.

3. Full abbreviations must be mentioned in the whole MS, for example, in line 68 names of the genes must be described for the readers.

4. The methodological part in the abstract should be modified accordingly in my opinion.

5. Histopathological image is not clear, it’s not possible to see clearly in the small panel. HE in the figure is without the description!!

6. Storyline missing in all of the figures!

7. Please make sure Figure 2E will be a higher resolution and proper pixels, can’t distinguish the proper differences. Also, I would suggest that graphs can be prepared in a colour-blind template.

8. Conclusion can be represented in a clearer form. 

Reviewer 3 Report

The authors of the study titled “Single-cell RNA Sequencing of the testis of Ciona intestinalis reveals the dynamic transcriptional profile of spermatogenesis in protochordates” explored the transcriptional profile of spermatogenesis in Ciona intestinalis using single-cell RNA sequencing. The study also involved the evolutionary conservation and diversity of spermatogenesis in different chordates and Homo sapiens.

I have carefully read the manuscript and have found it interesting and well-written and discussed.

I almost do not have concerns or comments and looking forward to seeing this manuscript published.

Specific points:

1-     In Materials and Methods: “H&E Staining” should be moved to a suitable place.

Author Response

Thanks for your kind comments. It has been modified according to your comments " In Materials and Methods: “H&E Staining” should be moved to a suitable place.". In addition, some tense errors and inappropriate statements were corrected in the manuscript.